# Diffusion-programmed catalysis in nanoporous material

Suvendu Panda[1], Tanmoy Maity [1,2], Susmita Sarkar[1], Arun Kumar Manna[1], Jagannath Mondal[1] & Ritesh Haldar [1] ✉

In the realm of heterogeneous catalysis, the diffusion of reactants into catalytically active sites stands as a pivotal determinant influencing both turnover frequency and geometric selectivity in product formation. While accelerated diffusion of reactants can elevate reaction rates, it often entails a compromise in geometric selectivity. Porous catalysts, including metal-organic and covalent organic frameworks, confront formidable obstacles in regulating reactant diffusion rates. Consequently, the chemical functionality of the catalysts typically governs turnover frequency and geometric selectivity. This study presents an approach harnessing diffusion length to achieve improved selectivity and manipulation of reactant-active site residence time at active sites to augment reaction kinetics. Through the deployment of a thin film composed of a porous metal-organic framework catalyst, we illustrate how programming reactant diffusion within a cross-flow microfluidic catalytic reactor can concurrently amplify turnover frequency (exceeding 1000-fold) and enhance geometric selectivity (~2-fold) relative to conventional nano/microcrystals of catalyst in one-pot reactor. This diffusion-programed strategy represents a robust solution to surmount the constraints imposed by bulk nano/microcrystals of catalysts, marking advancement in the design of porous catalyst-driven organic reactions.

Porous heterogeneous catalysts, like zeolites, metal-organic frameworks (MOFs) and covalent-organic frameworks (COFs) are widely researched for organic conversions[1-3]. The growing interest particularly in MOFs, constructed by linking metal ion/cluster with functionalized organic ligand, stems from their well-organized confined active sites, which are sterically and electronically tunable[4-6]. The extensive chemical diversity and predictable structure-property relationships of MOF-based catalysts have significantly expanded their applications, including in the areas such as enantiomer/size-selective catalysis, electrocatalysis, and photocatalysis[7-15]. Nonetheless, a persistent challenge for porous catalysts is the limitation imposed by mass transfer or diffusion[16,17]. Often all the catalytic sites are not accessible due to substantial diffusion barrier (surface barrier and large diffusion

length), leading to lowering of conversion efficiency[16]. Further, poorly regulated diffusion leads to smaller geometric selectivity. Therefore the overall efficiency of porous catalysts remains underestimated. Although the diffusion mechanism[18-20] is well-explored in theory and simulation for porous materials, access to the highest efficiency, selective catalysis in practice remains a complex challenge.

In typical catalytic reaction, whether batch or fixed-bed types, a range of MOF particle sizes is utilized. This size heterogeneity can significantly reduce both the turnover frequency (TOF) and geometric selectivity (which is the hallmark feature of MOF catalysis[21-23]). The underlying reasons are as follows: in porous heterogeneous catalysts, active sites are located both on the surface and within the bulk. When the catalyst has a very high surface-to-volume ratio (particle size < 10

[1]Tata Institute of Fundamental Research Hyderabad, Gopanpally, Hyderabad 500046 Telangana, India. [2]Present address: Haldia Institute of Technology, Department of Applied Science and Humanities, Hatiberia, ICARE Complex, Haldia, Purba Medinipur, West Bengal 721657, India. ✉e-mail: riteshhaldar@tifrh.res.in

nm), the reaction rate is governed by the surface active sites. For catalysts with a very low surface-to-volume ratio (particle size >10 μm), the very high reactant diffusion barrier allows the reaction only at the surface active sites. Within the intermediate particle size range, the diffusion rate controls the reaction outcome, influencing both TOF and selectivity. This relationship is depicted in Fig. 1a, b, demonstrating that for a specific catalytic reaction, there is a critical particle size or diffusion length ($L_c$) at which geometric selectivity is maximum (i.e. selectivity is controlled by diffusivity only). However, achieving this critical point in a conventional catalytic reaction appears unattainable. It is also noteworthy that maximizing geometric selectivity inherently leads to reduction in TOF; a trade-off intrinsic to any porous material-based catalytic reaction. Note that the diffusion length ($L_D$) is the maximum distance of the active site from the catalyst particle surface; hence it is shown as the radius of spherical particle. The rate of diffusion will vary depending on the reactant geometry, catalyst pore size, dimensionality and chemical functionality. In Fig. 1b, reactants of different sizes can fit into the pores of the catalysts and diffuse at different rate.

There are elegant chemical pathways (defect density control[24–27], postsynthetic linker modification, exchange[28–30], and multivariate assembly[31,32]) that can uplift TOF or geometric selectivity in MOF catalysis. For other types of porous catalysts (e.g. zeolites), the chemical tunability is not always straightforward. Keeping the chemical compositions constant in a MOF catalyst, we have explored the possibility of efficiency (TOF and selectivity) tuning by controlling the diffusion length only. We realized that the primary factors are diffusion length ($L_D$) and reactant-active site residence time (collision frequency). The former can improve geometric selectivity, while the latter can boost TOF. To precisely manipulate $L_D$, we have utilized a monolithic thin film of MOF catalyst with programmable thickness, and mounted in a

microfluidic cell (cross-flow) to enhance residence time. This reaction setup is illustrated in Fig. 1c (i). By precisely controlling the $L_D$ and residence time, for a condensation reaction we have achieved a >1000 fold increase in TOF and 2-fold enhanced geometric selectivity compared to a chemically equivalent batch reaction using submicron-size particles of catalyst. A straightforward control over $L_D$ and residence time allows programming the diffusion of the reactants, which controls the outcome of the reaction, as depicted in Fig. 1c (ii). In the following discussion we have illustrated the working principle and a general strategy to perform the diffusion-programmed catalysis, enabling access to the highest limits of TOF and geometric selectivity for any porous catalyst.

## Results

The pore window size, cavity size and chemical environment of MOFs can be engineered precisely and predictively[33–36]. By leveraging this chemical and structural tunability, catalytic sites within the pores can be tailored to accommodate specific reactants. This exceptional selectivity in geometry can be regulated by varying the $L_D$ of the reactants. Literature suggests that reducing $L_D$ (i.e., using smaller particle sizes) enhances the turnover frequency (TOF)[37–39]. However, this increase in TOF comes at the cost of reduced geometric selectivity. To evaluate this statement, we conducted a conventional one-pot reaction using submicron-scale catalyst particles.

We investigated a Knoevenagel condensation reaction catalyzed by a Lewis base (-NH₂) housed within a robust MOF pore[40,41]. The MOF of interest, UiO-66-NH₂[42], composed of Zr⁴⁺ and 2-aminobenzendicarboxylic acid (NH₂-bdc), was synthesized, with particle sizes[43] averaging ~ 500 nm and ~160 nm employed for the catalytic reactions (Supplementary Fig. 1–4). The MOF's triangular pore window size is ~6 Å and cavity diameter is ~11 Å (Supplementary

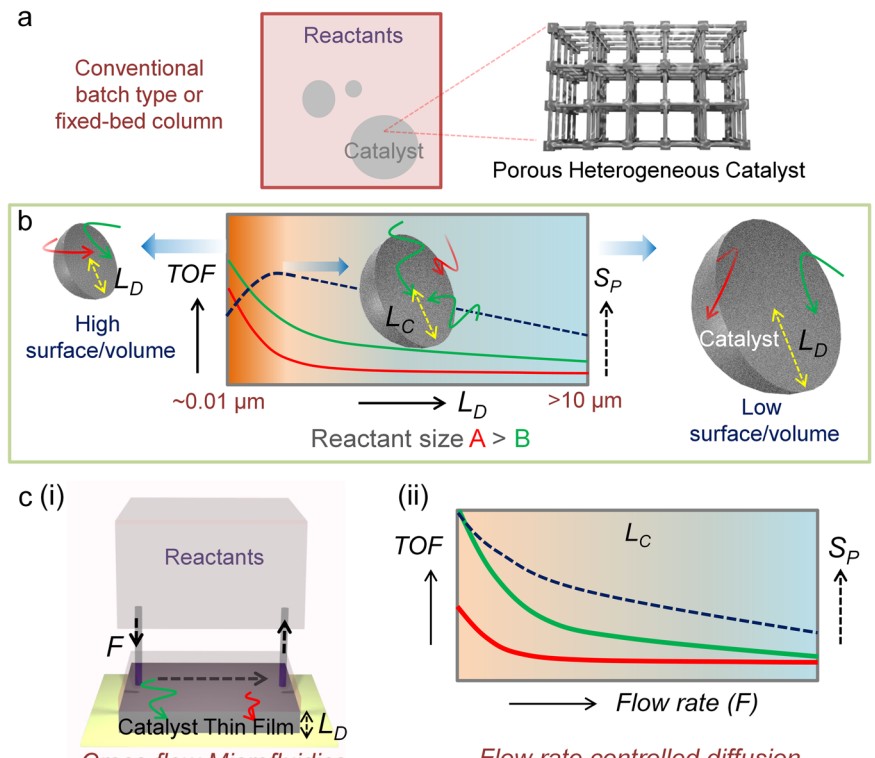

**Fig. 1 | Diffusion-programmed catalysis. a** Schematic illustration of porous heterogeneous catalyst (MOF) in a conventional reaction set up, **b** diffusion length-dependent turn-over frequency (TOF, solid line) and geometric selectivity (dotted line) profiles for a catalytic reaction; red and green arrow indicate diffusion for A and B reactants, size A > B, the half spheres represent a model porous catalyst particle, **c** (i) schematic illustration of the diffusion controlled, microfluidic cross-flow reaction set up, (ii) for a catalyst with $L_C$: plausible flow-rate ($F$) dependent TOF and geometric selectivity profiles, red and green line indicate reactants of different sizes. $S_P$ product selectivity. Following are the power laws for (**b**) and (**c**) (ii); TOF∝ $1/L_D^3$ and TOF∝ $1/F^{0.5}$.

Fig. 1a, b, experimentally obtained pores size distribution ~ 5 Å and 10 Å, due to structural defect, see later). Ethyl cyanoacetate (Et-CA; 4.5 Å × 10.3 Å) and tert-butyl cyanoacetate (t-But-CA; 5.8 Å × 10.3 Å) were chosen as the small and large nucleophiles[40], respectively, reacting with benzaldehyde (6 Å × 4.3 Å) to form ethyl-2-cyano-3phenylacrylate (Et-Acr) and ethyl-2-cyano-3phenylacrylate (t-But-Acr), respectively (Fig. 2a, Supplementary Figs. 5–7). The nucleophiles' dimensions allow diffusion into the pores, albeit at varying rates due to sieving effects. Molecular dynamics (MD) simulations were employed to estimate diffusivity differences of two

nucleophiles within UiO-66-NH$_2$, Supplementary Figs. 8 and 9 (see "Methods" section below). Analysis of mean square displacement (MSD) profiles indicates that Et-CA diffuses approximately ten times faster than t-But-CA (Supplementary Fig. 8). We have also employed biased simulation techniques, specifically umbrella sampling, to estimate the free energy barrier associated with molecular transport across the pore window (Supplementary Fig. 9a). Supplementary Figure 9b indicates that the free energy barrier for t-But-CA is higher than that for Et-CA, likely due to its larger size, suggesting slower pore to pore diffusion of t-But-CA compared to Et-CA. Close-

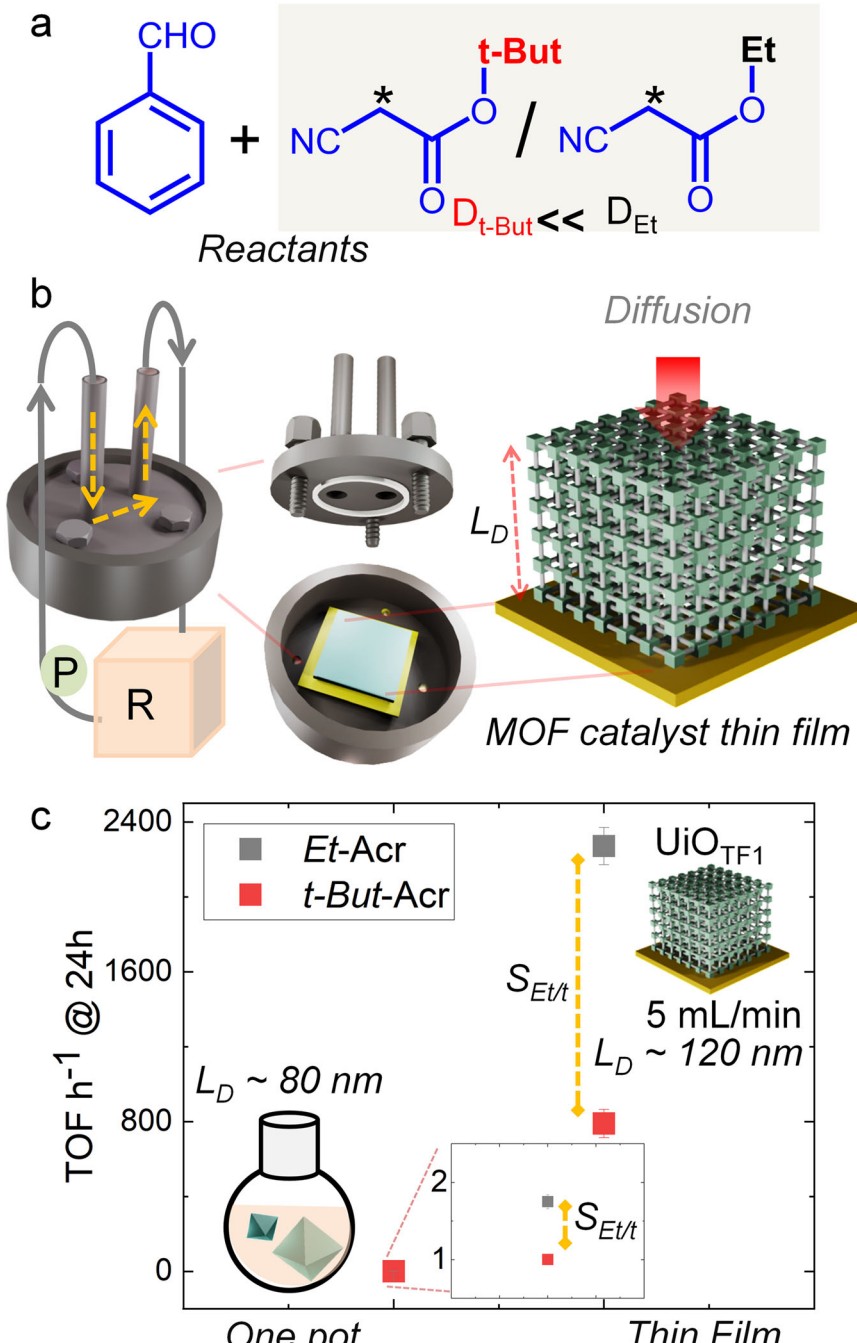

**Fig. 2 | Cross-flow microfluidic catalysis. a** Chemical structure of the reactants, * = reactive −CH$_2$ group; $D_{tBut}$ and $D_{Et}$ are the diffusivities of t-But-CA and Et-CA, respectively; **b** illustration of cross-flow microfluidic reaction using MOF thin film as catalyst, P circulating pump, R reactant and product chamber, yellow dotted line indicates the flow path, right MOF catalyst thin film with controllable thickness, $L_D$ diffusion length; **c** TOF after 24 h at 70 °C for one pot reaction with smaller MOF submicron-sized particles and in cross-flow microfluidic reaction using UiO$_{TF1}$ (5 mL/min flow-rate), 1 mmol of benzaldehyde and nucleophiles each in 10 mL of ethanol. Conversion % is calculated using $^1$H NMR.

proximity interactions between nucleophiles and MOF pores reveal that the active $-CH_2$ group (marked with * in Fig. 2a) of nucleophiles predominantly interacts with the organic linker rather than the metal node (Supplementary Fig. 9c, d). This $-CH_2-NH_2$ interaction suggests the potential for Lewis base catalyzed reactions.

We anticipated the reaction rate to exceed the diffusion rate, leading to diffusion-controlled TOF and selectivity. At 24 h, we achieved 42 ($\pm$2)% and 24% conversions for the smaller and larger nucleophiles, respectively, using the ~160 nm catalyst particles (1 mol% in ethanol, 70 °C; Supplementary Table 1, Supplementary Figs. 10–12). Under similar conditions, the ~500 nm catalyst particles exhibited approximately 50% lower yield but improved selectivity (~25%) for the smaller nucleophile. However, this improvement in selectivity is not deemed significant. Noteworthy, the one-pot reaction contains a large variety of particle sizes (Supplementary Figs. 2 and 3). Hence any significant change in the selectivity by varying the particle sizes is a challenge.

At this point, to regulate TOF and selectivity, we have designed a new reaction methodology. To enhance the TOF, we have done following: (i) a method to improve TOF is by increasing collision frequency (reactant-active site). To achieve this, we designed a microfluidic reactor where the catalyst is supported on a solid surface (*vide infra*) and the volume of reactant solution in contact with the catalyst is 80 ($\pm$4) µL. The small volume ensures that only a limited amount of reactant interacts with the entire catalyst layer at any given time, thereby enhancing reactivity. (ii) The microfluidic cell is connected to a pump that circulates a larger volume of reactant solution. Consequently, both reactants and the formed products circulate in a cross-flow direction along the catalyst bed (thin film). This reaction setup resembles one-pot catalysis reactions, except that the catalyst is placed within a microfluidic cell as a thin film. iii) Reactant flow can be controlled (0.1–15 mL/min) in a cross-flow direction, which helps preventing pore surface blockage.

To enhance the selectivity, the catalyst is deposited as a monolithic thin film with controllable thickness (see "Experimental" section, Supplementary Fig. 13). In the proposed cross-flow setup, concentration gradient is along the film thickness, and hence the thickness is $L_D$. This allows straightforward tuning of the $L_D$, unlike in the one-pot reaction (*vide supra*). The reaction setup is illustrated in Fig. 2b. Noteworthy those conventional nano/microparticles of MOFs are not suitable for this catalysis reaction. Rather, recently developed MOF thin film growth methodologies, e.g., layer-by-layer epitaxy[44–46],

chemical vapor deposition[47], solution atomic layer deposition[48], vapor assisted conversion[49], and electrochemical[50,51] deposition can be applied to make the catalyst layer.

To execute the reaction scheme, we have synthesized UiO-66-NH$_2$ monolithic thin film (UiO$_{TF1}$ - 120 nm thickness) on a Si/SiO$_2$ substrate at room temperature (298 $\pm$ 3 K), using a drop casting methodology[52] (Supplementary Figs. 14 and 15). This method is suitable for controlling the film thickness with high crystalline orientation (Supplementary Fig. 16). All the films synthesized have similar crystallite sizes (Supplementary Fig. 15) and crystalline orientations (Supplementary Fig. 17). Additionally, the oriented MOF thin film showed rapid adsorption of methanol[52] and tert-butanol vapors, confirming its porosity (Supplementary Fig. 18). Using the catalyst thin film UiO$_{TF1}$ we performed reactions using a similar reactant solution volume as for one-pot reactions, at 70 °C using a flow rate of 5 mL/min (Fig. 2b). We could realize maximum ~96% and 35% formation for the *Et*-Acr and *t-But*-Acr after 24 h (TOF ~ 2271 and 790 h$^{-1}$ respectively, Supplementary Table 2, Supplementary Figs. 19–21). The TOF and selectivity are enhanced by ~1300-fold and 1.5-fold, compared submicron-sized particle based catalysis. These findings, depicted in Fig. 2c, validate the simultaneous enhancement of TOF and selectivity through our proposed strategy. In absence of any chemical modification, these observations are unprecedented[17]. Subsequently, we elaborate on the underlying principles driving these radical improvements.

To confirm that the reaction is indeed controlled by pore diffusion and that selectivity enhancement is attributable the diffusion length ($L_D$), we have conducted reactions varying film thickness. Increasing film thickness simultaneously increases $L_D$ and also amount of active sites. In the absence of diffusion control, the TOF should remain unchanged with increasing thickness. Conversely, a diffusion-regulated reaction should exhibit a quadratically decreasing TOF, assuming the nucleophiles obey Fickian diffusion (TOF $\propto 1/L_D^2$). We synthesized UiO$_{TF2}$ and UiO$_{TF3}$ with thickness ~300 ($\pm$30) and 400 ($\pm$50) nm, respectively (Supplementary Fig. 14). The film TOF vs film thickness profiles for both of the condensation products are shown in Fig. 3a (Supplementary Table 2, under similar reaction conditions). The data clearly show that with increasing thickness TOF decreases and selectivity increases in a nonlinear trend. This supports the notion that the conversion is controlled by intrapore diffusion rather than being surface confined. Furthermore, it is feasible to enhance selectivity by adjusting the film thickness, i.e., $L_D$.

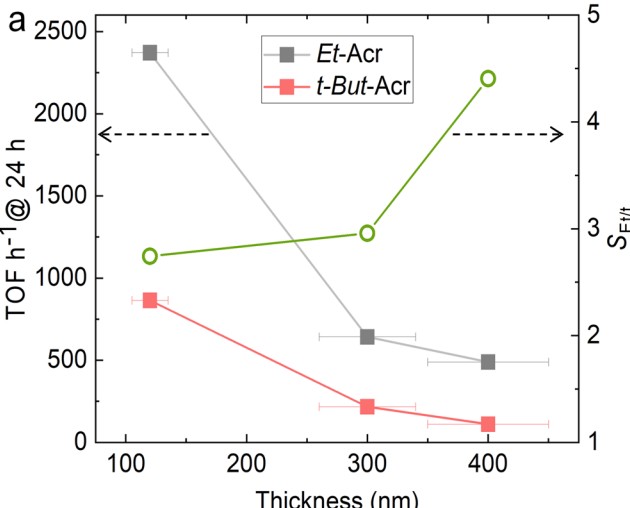

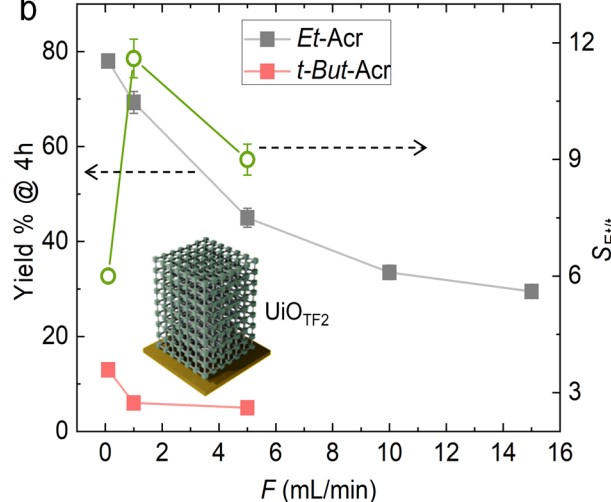

**Fig. 3 | Tunable TOF and selectivity. a** Catalyst film thickness dependent and (**b**) flow-rate (*F* mL/min) dependent TOF and selectivity ($S_{Et/t}$) profiles for the *Et*-Acr and *t-But*-Acr in cross-flow microfluidic reaction. All reactions are done at 70 °C using 1 mmol of benzaldehyde and 1 mmol of nucleophiles each in 10 mL ethanol. Conversion % is calculated using $^1$H NMR. Error-bars are calculated by carrying out three set of reactions.

Apart from selectivity, the sharp enhancement in TOF can be attributed to the microfluidic reaction set up. Direct evidence of this can be obtained by following experiment: A reaction of *Et*-CA and benzaldehyde was carried out in one-pot method using the thin film catalysts UiO$_{TF2}$. After 30 h the conversion was ~50 % (Supplementary Table 3, Supplementary Fig. 23). This markedly lower conversion efficiency compared to the microfluidic reaction, confirms that the effective residence volume for the reactant solution and entire catalyst indeed enhances the TOF.

To verify influence of other factors, e.g. defect density in the powder and monolithic thin films (UiOTF$_{1-3}$), we compared the infrared (IR) spectra, nuclear magnetic resonance spectrum (NMR) and X-ray photoelectron spectra (XPS) of those (Supplementary Figs. 24 and 25). The IR spectra indicated that the –COO stretching frequencies (asymmetric and symmetric 1571 and 1382 cm$^{-1}$, respectively) are similar for thin film and powder catalysts. XPS confirmed that the Zr/N ratio and nature of defects related to dangling –COO are similar[53,54] for both the type of catalysts. From solution state $^1$H-NMR of the disintegrated MOF powder we estimated ~33% missing linker defect (Supplementary Fig. 24b). We have also realized that monodispersed surface-anchored MOF particles can also enhance TOF[10,55]. We have confirmed that the UiO$_{TF1}$ is a monolithic thin film, having no evident cracks or islands of crystals (Supplementary Fig. 14a). These above mentioned evidences support that the enhanced TOF is due to the integration of MOFs in a cross-flow microfluidic setup.

After confirming the influence of diffusion and microfluidic reaction set up on the selectivity and TOF, respectively we have performed a flow-rate dependent catalysis reaction. Flow-rate and reactant-catalyst residence time is inversely proportional. In the absence of diffusion control, it is expected that product yield % will linearly decrease with increasing flow-rate. In case of diffusion regulated process, a quadratic decrease of conversion % is expected (*vide supra*). In Fig. 3b, *Et*-Acr yield % vs flow-rate profile (4 h reaction at 70 °C) exhibited a nonlinear trend (yield % $\propto (1/F)^{0.2-0.32}$), confirming reactant diffusion limited reaction (Supplementary Table 4, Supplementary Figs. 26 and 27). Moreover, *t-But*-Acr yield % also reduces nonlinearly (Supplementary Table S4, Supplementary Fig. 28) with increasing flow rate and we could observe highest selectivity of 11.5 at a flow-rate of 1 mL/min (*Et*-Acr yield ~70%). Using similar conditions (solvent, temperature, and concentration) in a one-pot reaction similar TOF and selectivity is not feasible. Noteworthy, the TOF and selectivity vs flow-rate profiles in Fig. 3b resemble Fig. 1b, except diffusion length is replaced by flow-rate. We have shown that by straightforward tuning of flow-rate it is feasible to achieve critical diffusion path ($L_c$) which allows access to highest selectivity with higher TOF than the conventional methods.

It is evident that for thickness and flow-rate based diffusion control experiments conventional Fickian diffusion is not followed, as the power law diverges from the quadratic norm. This anomalous behavior is indicative of additional factors, e.g., presence of competitive interaction with benzaldehyde and any specific chemical interaction with MOF. These factors inhibit the long-range random walk, leading to a modified TOF expressed as TOF $\propto (1/F)^{1/(2+\alpha)}$, where α represents the anomalous factor[56–58].

The above discussed experiments conclude that the TOF and selectivity can be enhanced beyond the conventional limits using the proposed scheme. A comparison of the TOF and selectivity, for similar condition (at similar temperature, solvent and time) one-pot (using 500 nm particle) and cross-flow microfluidic (using UiO$_{TF2}$) reaction, confirmed >1000 and 2-fold enhancement, respectively (Supplementary Tables 4 and 5, Supplementary Fig. 29). In the next step, we have explored the impact of heterogeneous mixtures (i.e., mixture of *Et*-CA and *t-But*-CA) on the reactivity. In Fig. 4 we have illustrated the observed TOF and selectivity for one-pot (1 mol% catalyst) and cross-flow microfluidic (4.2 × 10$^{-3}$ mol% catalyst, 0.1 mL/min) reactions (Supplementary Tables 6 and 7, Supplementary Figs. 30 and 31). It is

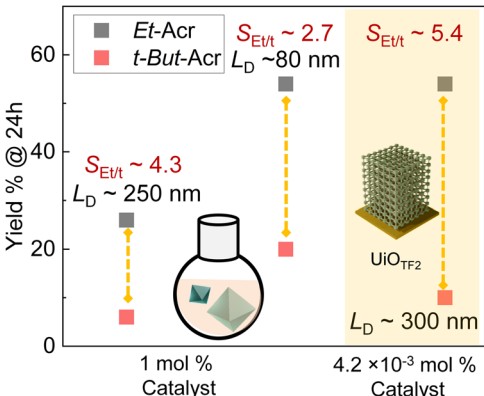

**Fig. 4 | Competitive diffusion.** Yield % and selectivity of the catalysis reactions using mixture (1:1 moles/moles) of nucleophiles. Conversion % is calculated using $^1$H NMR.

evident that competing diffusion of the reactants decreases the total conversion efficiency. However, the selectivity are found to be 5.4, higher than the one-pot reactions using large (4.3) submicron-sized catalyst particles. The improvement in the conversion efficiency is also evident; *Et*-Acr yield % is similar for one-pot (smaller particle) and cross-flow microfluidics methods, although catalyst mol% differ by >1000-fold. This confirms that even in a heterogeneous mixture of competing reactant diffusion, cross-flow microfluidic reaction using MOF catalyst monolith is superior to the state-of-the-art catalysis reactions while maintaining the crystallinity of the monolithic thin film post-reaction (Supplementary Fig. 16).

## Discussion

Metal−organic frameworks (MOFs) are esteemed as exceptional heterogeneous catalysts among the porous materials, owing to their crystallinity, extensive chemical versatility, high surface area, and confinement effects. However, the diffusion-limited TOF and selectivity present significant challenges that continue to impede their catalytic performance. To address these issues, we have engineered a cross-flow microfluidic reaction setup and catalyst monolithic thin film, which allow precise control over reactant diffusion, thereby modulating both TOF and selectivity. This diffusion programmability is exemplified in a Knoevenagel condensation reaction, demonstrating the feasibility of simultaneously enhancing TOF and product selectivity beyond the conventional limits imposed by one-pot or batch reactors, achieving improvements by orders of magnitude. The ability to control diffusion rate allows for continuous tuning of the conversion efficiency, with experiments revealing a sub-diffusive nature of the catalytic reaction. This marks the first proof-of-concept demonstration of diffusion-programmed catalysis in porous materials, representing a significant step forward in the field. The design of microfluidic cell and flow control used in the current work may be improved in future to explore other varieties of organic reactions.

## Methods
### Characterization techniques
**X-ray diffraction.** The X-ray diffraction patterns of the powder and thin films were recorded using a Rigaku SmartLab diffractometer using nickel-filtered Cu *K*α radiation (λ = 1.5418 Å). Data were collected from 5 to 20° at room temperature (voltage 40 kV, current 200 mA). XRD was recorded in 2θ/θ (step size 0.01, scan rate 0.2°/s) geometry.

**Scanning electron microscopy (SEM).** Morphology and cross-section of the thin films and powders were characterized using field emission scanning electron microscopy (FESEM), JEOL JSM-7200F instrument with a cold emission gun operating at 5, 25, and 30 kV.

**Infrared (IR) spectroscopy.** Infrared reflection absorption spectroscopy (IRRAS) of the thin films and attenuated total reflection (ATR) absorption spectroscopy of the powder were done using the Bruker Vertex 70 v instrument, with a spectral resolution of 2 cm⁻¹. IRRAS results were recorded in grazing incidence reflection mode at an angle of incidence 45° relative to the surface, under vacuum at room temperature. 1-octadecanethiol self-assembled monolayer (SAM) functionalized Au was used for background measurements.

**X-ray photoelectron spectroscopy (XPS).** Elemental detection of the powder and thin film was performed using X-ray photoelectron spectrometer (PHI versaProbe III) within an ultrahigh vacuum ($1 \times 10^{-9}$ bar) environment. This instrument was equipped with an Al-$K\alpha$ X-ray source and a monochromator.

**Nuclear magnetic resonance (NMR) spectroscopy.** NMR spectra were recorded on a BrukerNanoBay 300 MHz NMR spectrometer. In our experiment, after the reaction, we used a syringe filter to separate the MOF powder catalyst. The entire collected solution was then concentrated and 0.4 mL of $CDCl_3$ was added to check the NMR data. For the cross-flow microfluidic catalysis, we collected the entire solution from the fluidic cell, concentrated and after adding 0.4 mL of $CDCl_3$ we checked the NMR data.

**Analysis of mass uptake kinetics.** We evaluated the mass uptake rates of thin films grown on quartz crystal microbalance (QCM) sensors functionalized with a MUD-coated gold surface. To probe the system, we used methanol and *tert*-butanol, under 50 ml/min $N_2$ flow) as test molecules, chosen for their smaller kinetic diameters relative to the pore window size. The QCM sensors with thin-film coatings were mounted within a fluidic cell in a temperature-controlled setup, where vapor uptake rates were measured by monitoring changes in the fundamental frequency over time.

The relationship between mass and frequency for the QCM sensor is governed by the Sauerbrey equation:

$$\Delta m = - c \frac{\Delta f}{n} \qquad (1)$$

Here $n$ represents the overtone order (specifically $n = 3, 5,$ and $7$) and $c$ stands for the mass sensitivity constant. For a quartz crystal with a frequency of 5 MHz, the value of $c$ is 17.7 ng cm⁻². We analyzed the data under the assumption of Fickian diffusion, which means we considered a constant diffusivity, $D$, that does not change with varying vapour concentrations.

**Molecular dynamic simulation of reactant diffusion**
**Simulation model.** We considered a $1 \times 1 \times 1$ UiO-66-$NH_2$ containing one molecule of *Et*-CA or *t-But*-CA individually. Partial charges for the MOF atoms were obtained from earlier report[59], and other bond, angle, and dihedral parameters were modeled using OBGMX. The analyte molecules were modeled using Charmm force field generic parameters (CGenFF)[60]. Simulations for each reactant molecule were conducted in the gas phase, within a rectangular box of dimensions $2.07 \times 2.07 \times 2.07$ nm³.

**Simulation method.** Each simulation employed periodic boundary conditions (PBC) in all three dimensions. Long-range electrostatic interactions were managed using the particle mesh Ewald (PME) method[61] with cubic interpolation, with a 1.2 nm cutoff for short-range electrostatic interactions. The LINCS algorithm[62] was used to constrain bonds involving hydrogen atoms. The system was first energy minimized using the steepest descent algorithm, followed by stepwise equilibration over seven steps with gradual temperature increases

from 50 K to 300 K, each step lasting 100 ps with a time step of 0.0005 ps. During equilibration, the average temperature was maintained using a V-rescale thermostat[63], separately coupling the MOF and analyte molecule. The equilibrated system then underwent a 10 ns NVT production run at 300 K, also maintained by the V-rescale thermostat. All simulations were performed using GROMACS[64] version 2018.6 and repeated multiple times to ensure statistical reproducibility. Structure files are provided as UiO-Et and UiO-tBut in pdb format.

Reactant diffusion was analyzed by calculating the mean square displacement (msd) using the 'gmx msd' tool, and the corresponding diffusion coefficient ($D$) was estimated. To understand the chemical interactions between reactant molecules and the MOF, pair correlation functions of active $-CH_2$ groups (see Fig. 2a) were measured relative to specific functionalities in MOF (metal-oxo nodes, organic linkers, and $-NH_2$ group of the organic linkers). Metal-node represents the Zr metal and the linked oxygen atoms, $\mu_3$-O and $\mu_3$-OH. Whereas the term linker represents all the atoms of the organic linker, i.e., bdc-$NH_2$.

**Free energy calculation.** To investigate the free energy barrier associated with molecular movement across the MOF pores, we conducted umbrella sampling simulations. These simulations focused on evaluating the energetics of molecular motion across the MOF pore by simulating the movement of a single molecule from one pore window to another. The equilibrated structure from prior simulations was used as the starting point, with the dimensions of the simulation box and MOF remaining unchanged.

Initial configurations for each umbrella sampling window were generated by systematically moving the molecule along the Y-direction through the MOF pore, from one pore window to the next. The Y-direction of the molecule within the pore of the MOF ($d$) was chosen as the collective variable (CV) for the umbrella sampling. Harmonic potentials with force constants in the range of 7500 kJ/mol/nm² were found to be optimal for CV values spanning from 0.05 nm to 0.85 nm (with a spacing of 0.05 nm), ensuring sufficient overlap between adjacent windows.

For each umbrella sampling window, the same simulation protocol was applied. The Weighted Histogram Analysis Method (WHAM) [Grossfield, A. WHAM: The Weighted Histogram Analysis Method version 2.0.7, http://membrane.urmc.rochester.edu/content/wham] was then used to analyze the simulation data. WHAM enabled the calculation of unbiased histograms and the corresponding free energy profiles by combining the results from all independent trajectories.

Umbrella sampling simulations were performed for both substrates, *Et*-CA and *t-But*-CA, and the resulting free energy profiles were (Fig. S9b) compared to elucidate differences in their molecular transport through the MOF pore.

## Data availability
The data that support the findings of this study are available from the corresponding authors upon request. Source data are provided with this paper.

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

## Acknowledgements

We acknowledge the intramural funding at TIFR Hyderabad from the Department of Atomic Energy (DAE), India, under Project Identification Number RTI 4007.

## Author contributions

S.P. and R.H. conceived the idea and planned the experiments, S.P. performed the experiments, characterization, and analysis with gui-dance from R.H., T.M., and A.K.M. carried out parts of the synthesis, S.S. performed the MD simulation with guidance from J.M., paper draft was prepared with the inputs from all the authors.

## Competing interests

The authors declare no competing interests.
