## [Peer review file · Nature Communications]

Diffusion-programmed catalysis in nanoporous material

Corresponding Author: Dr Ritesh Haldar

Version 0:

Reviewer comments:

Reviewer #1

(Remarks to the Author)

The authors have beautifully demonstrated the concept of diffusion programmed catalysis in porous materials by using a MOF-based model system and employing a cross-flow microfluidic reactor simulating a typical batch-type heterogeneous (solid/liquid) catalysis experiment. By precisely controlling the MOF catalyst film thickness and as well carefully ruling out adverse effects or intrinsic system issues such as the importance of MOF-typical defects, the authors could verify the expected relationships of TOF and selectivity on the diffusion length $L(D)$ and flow rate F . Congratulations to these nice results. The level of technical quality of the manuscript is very high. If there are any aspects that should be considered before acceptance, it may be worth to substantiate the characterization of the MOF thin film materials work (section 2.8, SI). A lot of convincing data are provided for the MOF particle synthesis as the reference system and the catalysis work, analysis in terms of diffusion and flow rate modulation and so on, the key material, however, are the MOF thin films of various thickness and defined quality. Here, the provided data are quite short: Figs S13, S14. The variation of the film thickness Fig S13 a-d is shown, however the film quality is not sufficiently well documented. Also, the main text/discussion does not provide rigorous data in terms of the primary MOF particle sizes present in the films. Clearly, the films are composed of aggregated MOF particles of undefined size. Also, film morphology in terms of density, homogeneity, crack-free, surface roughness etc. is not well documented/discussed. In particular UiO(TF1) of 120 nm is of poor quality - how can the authors come up with the 120 nm thickness given that very rough film morphology? The porosity of the films, i.e. a primary property of MOFs is not independently proven (e.g. by adsorption isotherms, obtained by QCM techniques or similar). Admittedly, the derived relationships and comparisons of the MOF films within the microfluidic device with the powder reference materials are convincing, however a more rigorous establishment of the MOF film quality is necessary (also a critical reflection of the dependence of the key measurements and conclusions on that film quality). At this point, the authors may consider that a drastic enhancement of TOF by downscaling MOF particles to the size of 100 nm (and below) together with surface anchoring of the particles (and avoiding or limiting MOF particle aggregation, thus intrinsically modulating diffusion limitations) has been demonstrated before (e.g.: <https://doi.org/10.1021/acscatal.0c00550>). The authors may give reference to such related work and differentiate their new concept properly from those previous findings. However, this referee thinks that the manuscript may be very well suited for acceptance and publication in Nature Communication if the authors have convincingly addressed these remaining issues.

Reviewer #2

(Remarks to the Author)

The authors showcase a microfluidic flow-cell system for carrying out catalytic reactions using thin films of porous membranes. The authors argue that controlling the thickness of the catalyst thin film and the flow rate of the reactor governs TOF and product selectivity.

I believe that attributing the changes in catalytic performance for different thin films to their differences in thickness (or particle size) needs to be further discussed and possibly deconvoluted from the differences in composition due to possible defects in the materials synthesized using modulators. The flow-rate-dependent results indicate that their results are indeed promising, yet one of my main criticisms is the possibility of the molecules to even enter the pores of the material, which has relatively small pore apertures. I believe these two main points, as well as the ones below, should be addressed. While I am not an expert in organic transformations, I believe that my comments would serve to strengthen the materials part of the paper.

The methodology of the paper is sound, but, as is stated above, I believe that some further investigations or explanations

regarding the composition (defects, surface vs. interior sites being catalytically active) need to be discussed.

1. In Figure 1B, the authors show a general proportionality that smaller L_d is proportional to a higher turn-over frequency. However, in Figure 1C, for the one-pot case, a smaller TOF is observed for the smaller particles. How do the authors consolidate this? I apologize if I have missed anything in the rationalization. Is the influence of the controlled flow the only changing factor here?

2. In row 47-49, the authors state that:

“When the catalyst has a very high surface-to-volume ratio (particle size <10 nm), the reaction rate is governed by the surface active sites. For catalysts with a very low surface-to-volume ratio (particle size >10 μm), the very high reactant diffusion barrier limits the reaction at the surface active sites.”

How does the large diffusion barrier serve to limit the reaction at surface sites? Would it not do the opposite?

3. Overall, is the diffusion length (L_d) meant as a mean-free path for substrates before reaching an active site? This might need some clarification. In Figure 1, is L_d simply drawn as the radius of the particle? Would L_d not also depend on the nature of the pores (aperture sizes, pore network dimensionality, spatial frequency of active sites)? Perhaps this could be elaborated? Along those lines, could their mechanistic rationale be put in more accessible terms that speak to the broader field of heterogeneous catalysis?

4. Would the authors expect the influence of film thickness to be significantly stronger if the catalyst film was used as a membrane (i.e. reaction mixture flowing through, rather than over the material)? Perhaps the authors could comment on this.

5. The relationship of L_d w.r.t to TOF and selectivity is then extended to apply onto film thicknesses, as a means of precisely controlling L_d . Could the authors comment on the potential influence of surface roughness, and how it seems to vary for the different samples?

6. On a similar note, there should be a flow rate during which film thickness should not influence the catalytic results, would it be possible to estimate this? This is perhaps similar to question number 9 below.

7. In Figure 1 and the related part of the introduction, are there any appropriate references that could be given for the fundamental relationship between diffusion length, TOF and selectivity? The caption of figure one also gives two proportionalities for how TOF varies with respect to diffusion length and flow rate. Are these established relationships?

8. It seems like an axis label is missing on Figure 2c, what is the x-axis?

9. What is the diffusion length under the studied conditions? Is there a point where the film thickness becomes irrelevant (i.e. diffusion into film does not occur past a certain point with a certain flow rate)?

10. What is the expected diffusion rate of the substrates inside the MOF? As I understand it, the argument is that a thicker film would give a longer diffusion length (L_d) when compared to a thinner film (assuming the same flow rate), why is that the case? This perhaps is closely related to question number 8 above.

11. Considering the pore aperture size of the framework (UiO-66-NH₂), would the authors expect the substrates to fit into the pores of the framework? How does the size of the pore aperture compare to the bulkiest groups of the substrates that are used? Can the substrates be expected to reach the interior of the film?

12. A common reason for enhanced catalytic activity in UiO-66 is the presence of various defects. Could the authors comment on any potential presence of defects in their materials? Particularly how their synthesis route would compare to typical syntheses of more or less defective Ui-66-NH₂? As an example, a range of missing-linker defect concentrations are seen in materials prepared with varied linker-to-modulator ratios (10.1021/acs.chemmater.6b00602, 10.1016/j.cej.2023.143394). Could this be a potential confounder for the observed differences in catalytic performance of their materials or thin films?

This should at the very least be critically discussed considering that the differences in catalytic activity is the main finding of the paper. One approach for deducing this, at least for the powdered samples, would be thermogravimetric analysis, showing the lower/higher mass loss as the linker is burned off in air (10.1021/acs.chemmater.6b00602). The current evidence presented by the authors, using XPS, is only sensitive of the nature of the exterior surface, and gives little information about the composition of the rest of the film. Even then, doing a quantitative analysis of XPS data requires the use of relevant standards as a comparison. The authors currently only discuss the composition of their materials as a comparison between particles vs. films, I believe this discussion of composition and defects needs to be extended to the films of various thicknesses (or particles of different sizes), as their defect concentrations may be radically different (and thereby giving rise to the differences in catalytic performance).

13. It would be nice to show that the thin films remain crystalline after the catalysis - they likely are, as Zr-MOFs are used for their high chemical robustness - but PXRD patterns taken also after catalysis should be provided.

14. On another note regarding PXRD measurements, is the instrument listed correct? I can not seem to find records of

Rigaku producing a PXRD instrument called "XDS 2000". Perhaps the manufacturer has been mixed up? Also, 200 mA seems to be a very high current for a conventional X-ray tube, but it may very well be correct, I would urge the authors to double-check.

Version 1:

Reviewer comments:

Reviewer #1

(Remarks to the Author)

The authors have done a very good job in addressing the numerous issues raised during review. The manuscript has significantly been strengthened by the new data/information added to the Supporting Information and by some specific sections/sentences to the main text / discussion, as explained in the response letter. The revisions are conclusive. I am happy to recommend acceptance of the manuscript as it is now.

Reviewer #2

(Remarks to the Author)

The authors have certainly made a great effort to address comments from the reviewers. I am grateful for the thorough explanations provided by the authors, in particular regarding the mechanistic understanding of the selectivity and diffusion processes, as well as the description of the diffusion length. After reading the response letter and the revised manuscript, I have listed my comments and questions below, which I believe should be addressed prior to acceptance. Most pressing is perhaps the proof of the substrates fitting into the pore apertures of the porous catalyst. While the QCM results give new insights, I believe the discussion would benefit from the results being put into context (i.e. how the uptake relates to the amount of catalyst).

1. In their second response to reviewer one, the authors state that the defect density has been accounted for. This is mainly inferred through XPS and IR data, yet a more definitive and quantitative measure would be any deviation in the stoichiometry of the materials as determined by e.g. thermogravimetric analysis. While not possible to carry out on the thin films, could the authors perhaps comment on the relative ratios between the different oxygen environments (Zr-O-Zr, Zr-O-C, COOH) for the different thin films as observed in the XPS results? Visually it appears to be similar between the thin films, as the authors highlight in the main text.
2. Additionally, regarding the XPS results and the defect concentration, the C-C and C-N binding energies differ for the thin films, and a handful of peaks in the XPS data are shifted when comparing the thin films vs. the small particle sample. Could the authors comment on what the possible differences are between the thin films and the small particles? Are the magnitudes of these shifts worth considering?
3. Regarding the new QCM results, would it be possible to give the numbers in specific uptake (ng/g_MOF or similar)? Could the authors possibly discuss the magnitude of the uptake? Also, at what temperature was the measurement carried out? Would it be possible to measure a blank sample to make sure that the uptake is solely due to the porous material? Could the uptake be occurring only at the surface for the tert-butanol case?
4. The MD simulations show displacements that are quite small. If I interpret the y-axis values correctly, the movement is only within a single pore. If this is correct, I do not believe it shows the feasibility of the bulkier substrates to pass through the pore aperture. Would it be possible to carry out simulations at longer timescales to show diffusion between pores? On this note, I agree that the substrates fit inside the pores, but I am more hesitant if they can pass through the aperture, which, when taking into account the van der Waals radii of the carbon atoms, has a limiting size of about $\sim 4 \text{ \AA}$ depending on the orientation of the aromatic rings on the organic linker molecule, and further assuming that all amine groups are pointed away from the pore.
5. Additionally, how were the distances defined for the CH₂-to-linker/cluster interactions? I.e. what atom on the cluster or linker was used? For the amino group distance (Figure S9c), was the H₂C - - - NH₂ distance used?
6. Also pertaining to the simulations (Figure S9), is it diffusion constant or diffusivity (diffusion coefficient)? Depending on the answer, is the unit correct? Are the authors certain about the number of significant digits in their results?
7. Lastly regarding the MD results, would it be possible to include the structure file in an electronic format rather than as lines in the supporting information? I believe this would greatly improve the ease of access to the data, and also greatly reduce the number of pages of the document.
8. Regarding the SEM results, some of the images is acquired at 5 kV and 25 kV, not only at 30 kV as is listed in the Methods section.
9. At line 118 the authors state that "We anticipated the reaction rate to exceed the diffusion rate". Does this mean, that under the studied conditions, the reaction rate is high enough so that the diffusion length does not become limiting?
10. This is merely this reviewer being picky, but perhaps "in heterogeneous catalysts" at line 45 should be changed to "in

porous heterogeneous catalysts”.

11. The authors also talk interchangeably about “monolith”, “thin film”, and monolithic thin film. Perhaps consistently using “monolithic thin film” would be preferable? I leave this to the discretion of the authors, I just wanted to raise the question.

12. Is there a reason why the NMR results are truncated to show shifts only between 8 to 10 ppm? Could the authors provide the full spectra and interpretation? Typically, a range from -1 to 9 ppm is recommended.

Version 2:

Reviewer comments:

Reviewer #2

(Remarks to the Author)

The authors have supplied answers to the comments of the reviewer's question. The conclusions are further supported by new experimental data and simulations. I believe the manuscript should be accepted in its revised form.

Point-by-point response

Reviewer #1:

The authors have beautifully demonstrated the concept of diffusion programmed catalysis in porous materials by using a MOF-based model system and employing a cross-flow microfluidic reactor simulating a typical batch-type heterogeneous (solid/liquid) catalysis experiment. By precisely controlling the MOF catalyst film thickness and as well carefully ruling out adverse effects or intrinsic system issues such as the importance of MOF-typical defects, the authors could verify the expected relationships of TOF and selectivity on the diffusion length $L(D)$ and flow rate F . Congratulations to these nice results. The level of technical quality of the manuscript is very high.

Response: We are glad to learn that the reviewer acknowledges the novelty of our work and recommends its publication.

If there are any aspects that should be considered before acceptance, it may be worth to substantiate the characterization of the MOF thin film materials work (section 2.8, SI).

Response: Thank you for these suggestions. In the revised version we have included additional experimental details and characterizations to validate the quality of the MOF thin films used for catalysis.

Note that the thin film fabrication methodology used in the current work has been independently developed recently by our group (<https://pubs.rsc.org/en/content/articlelanding/2024/sc/d3sc06719j>). Thin film growth mechanism, crystalline orientation, defect density, surface roughness and porosity, all of these aspects have been taken care of while using the thin films for catalysis. According to the suggestion, we have included all those relevant informations in the revised supporting information.

A lot of convincing data are provided for the MOF particle synthesis as the reference system and the catalysis work, analysis in terms of diffusion and flow rate modulation and so on, the key material, however, are the MOF thin films of various thickness and defined quality. Here, the provided data are quite short: Figs S13, S14. The variation of the film thickness Fig S13 a-d is shown, however the film quality is not sufficiently well documented. Also, the main text/discussion does not provide rigorous data in terms of the primary MOF particle sizes present in the films. Clearly, the films are composed of aggregated MOF particles of undefined size. Also, film morphology in terms of density, homogeneity, crack-free, surface roughness etc. is not well documented/discussed. In

particular UiO(TF1) of 120 nm is of poor quality - how can the authors come up with the 120 nm thickness given that very rough film morphology?

The porosity of the films, i.e. a primary property of MOFs is not independently proven (e.g. by adsorption isotherms, obtained by QCM techniques or similar).

Admittedly, the derived relationships and comparisons of the MOF films within the microfluidic device with the powder reference materials are convincing, however a more rigorous establishment of the MOF film quality is necessary (also a critical reflection of the dependence of the key measurements and conclusions on that film quality).

Response: Thank you for these suggestions.

We have worked on the thin film characterizations, and included those data in the revised manuscript. These changes are all highlighted in yellow.

About the particle size: We have checked the average particle sizes of the different MOF thin films (UiO_{TF1-3}) from SEM morphologies. Those are 120-150 nm sized and now added in the revised version (Figure S14).

We have also compared the full-width half maxima of the diffraction peaks from (111) plane, which indicated small difference for the three thin films (Figure S17). This confirms that particle size is not the governing factor; rather the film thickness controls the diffusion and TOF of the reaction.

About the film quality: We acknowledge that TF1 film looks rough in Figure S13. We have carefully checked the cross-section SEM, and now can confirm that the TF 1 thickness is 120 ± 10 nm (Figure S14).

Further, we have also compared the IRRAS of the thin films (Figure S24). The specific stretching frequencies of COO remain unaltered for the UiO_{TF1-3}. This confirms the thin films are compositionally similar.

We have carried out XPS analysis of the thin films UiO_{TF1-3}. The high resolution XPS also confirmed that the different thicknesses of thin films did not affect the composition. (Figure S25)

About the porosity of thin films: In our earlier report, we have assessed the porosity of the thin film (<https://pubs.rsc.org/en/content/articlelanding/2024/sc/d3sc06719j>). According to the suggestion, we measured tertiary butanol adsorption (kinetic diameter ~ 6.2 Å) of the UiO-66-NH₂ thin film using quartz crystal microbalance method. This is included in the revised version of the supporting information (Figure S18).

Please see the figures appended below:

Figure S14. a) SEM morphology image of UiO_{TF1} . SEM cross-section image of b) UiO_{TF1} , c) UiO_{TF2} , and d) UiO_{TF3} .

Figure S15. a-c) SEM morphology images and particle size distribution of UiO_{TF1-3} . Note, the cracks are due to beam exposure.

Figure S24: ATR and IRRA spectra of the submicron sized UiO-66-NH₂ crystallites and UiO_{TF1}-

Figure S25: XPS comparison of the UiO_{TF1-3} and 160 nm crystallites of UiO-66-NH₂. a) overview scan, b-e) high-resolution scan of C, Zr, O, and N respectively.

Figure S17: XRD patterns of the UiO_{TF1-3}, showing consistent full-width half maximum (FWHM) across all the three samples.

Figure S18. Total mass uptake profile of methanol and *tert*-butanol vapors, for UiO_{TF1} thin film using quartz crystal microbalance experiments.

At this point, the authors may consider that a drastic enhancement of TOF by downscaling MOF particles to the size of 100 nm (and below) together with surface anchoring of the particles (and avoiding or limiting MOF particle aggregation, thus intrinsically modulating diffusion limitations) has been demonstrated before (e.g.: <https://doi.org/10.1021/acscatal.0c00550>). The authors may give reference to such related work and differentiate their new concept properly from those previous findings.

Response: Thank you for bringing this published work to our attention. We have included this in the revised manuscript with a brief discussion, highlighted in yellow.

However, this referee thinks that the manuscript may be very well suited for acceptance and publication in Nature Communication if the authors have convincingly addressed these remaining issues.

Response: We again thank the referee for insightful suggestions and appreciation. We believe we have responded to queries and this version of the manuscript will be accepted for publication.

Reviewer #2:

The authors showcase a microfluidic flow-cell system for carrying out catalytic reactions using thin films of porous membranes. The authors argue that controlling the thickness of the catalyst thin film and the flow rate of the reactor governs TOF and product selectivity.

I believe that attributing the changes in catalytic performance for different thin films to their differences in thickness (or particle size) needs to be further discussed and possibly deconvoluted from the differences in composition due to possible defects in the materials synthesized using modulators.

Response: Thank you for this comment. We have deconvoluted the effects, and in the revised version we have highlighted this aspect clearly (also see the pointwise responses in the following).

Following points are in support of the proposed hypothesis of diffusion control:

- 1) The control over TOF and size selectivity is observed for different sized powder MOFs and different thickness MOF thin films. Different size and thickness will have differences in the active site and defect densities. However these differences are not significant enough. To support this fact, we have now compared IR and XPS data (see in the following responses and revised manuscript) of the different MOF particles and thin films.
- 2) For the same thin film, we have observed modulated TOF and selectivity by changing the flow-rate. This is not feasible if defect densities are the primary control factor.

The flow-rate-dependent results indicate that their results are indeed promising, yet one of my main criticisms is the possibility of the molecules to even enter the pores of the material, which has relatively small pore apertures. I believe these two main points, as well as the ones below, should be addressed.

Response: Thank you for this comment.

We are not sure whether we have understood the comment correctly. If not, kindly correct us.

According to our experimental observations, substrates can enter into the pores of the framework. This pore sizes are the reason of size-selective catalysis, as presented in Figure 2c, 3 and 4. We have checked the compatibility of the substrates size with pore size of the framework, and also performed additional experiment to confirm the permanent porosity of the MOF.

Figure S17 shows the tertiary butanol and methanol vapor uptake profiles, measured by quartz crystal microbalance experiment. Tertiary butanol molecular dimension is comparable to the larger reactant (*t-But-CA*) size used in this work.

We have also carried out MD simulation to probe the molecular diffusion in the UiO-66-NH₂ MOF. This is shown in Figure S8 and 9. It clearly showed that the reactants can fit into the pores and rate of diffusion is size dependent.

While I am not an expert in organic transformations, I believe that my comments would serve to strengthen the materials part of the paper.

Response: We thank the referee for insightful suggestions. In the following we have responded to each of the comments.

The methodology of the paper is sound, but, as is stated above, I believe that some further investigations or explanations regarding the composition (defects, surface vs. interior sites being catalytically active) need to be discussed.

Response: We have addressed the specific concerns and revised the manuscript accordingly.

1. In Figure 1B, the authors show a general proportionality that smaller L_d is proportional to a higher turn-over frequency. However, in Figure 1C, for the one-pot case, a smaller TOF is observed for the smaller particles. How do the authors consolidate this? I apologize if I have missed anything in the rationalization. Is the influence of the controlled flow the only changing factor here?

Response: We apologize for this confusing Figure description.

Figure 1c does not compare between different sized particles. This picture describes that how TOF and selectivity change with flow-rate for different sized reactants. The diffusion length remains constant.

2. In row 47-49, the authors state that: “When the catalyst has a very high surface-to-volume ratio (particle size <10 nm), the reaction rate is governed by the surface active sites. For catalysts with a very low surface-to-volume ratio (particle size >10 μm), the very high reactant diffusion barrier limits the reaction at the surface active sites.”

How does the large diffusion barrier serve to limit the reaction at surface sites? Would it not do the opposite?

Response: We apologize for this confusing sentence.

The corrected sentence is “the very high reactant diffusion barrier allows the reaction only at the surface active sites.”

3. Overall, is the diffusion length (L_d) meant as a mean-free path for substrates before reaching an active site? This might need some clarification. In Figure 1, is L_d simply drawn as the radius of the particle? Would L_d not also depend on the nature of the pores (aperture sizes, pore network dimensionality, spatial frequency of active sites)? Perhaps this could be elaborated?

Along those lines, could their mechanistic rationale be put in more accessible terms that speak to the broader field of heterogeneous catalysis?

Response: Thank you very much for these suggestions.

No, the diffusion length (L_D) described here is not related to the mean-free path. We define this as the maximum depth from the surface of the catalyst particle reactants diffuse to reach active site. Hence, for spherical particles, L_D is calculated as radius of the particle.

Changes in the nature of the pore size, chemical functionality, reactant size will not change the (L_D), but only change the diffusivity.

Note that our first hand clause is that the reactants can fit into the pore and can diffuse into the nanochannels of the catalyst. Hence, we define the diffusion length in terms of particle size.

We have now introduced a brief discussion to make the definitions more clear and highlighted in yellow.

4. Would the authors expect the influence of film thickness to be significantly stronger if the catalyst film was used as a membrane (i.e. reaction mixture flowing through, rather than over the material)? Perhaps the authors could comment on this.

Response: This is a very important point that we would like to address in near future.

At this point, we believe that the flux (moles/time) through a membrane will be influenced by the film thickness. Only difference will be that the diffusion length will be half of the membrane thickness.

However, the flow dependent TOF and selectivity will not be observed. This is because, the cross flow method controls the substrate-pore retention time.

5. The relationship of L_D w.r.t to TOF and selectivity is then extended to apply onto film thicknesses, as a means of precisely controlling L_D . Could the authors comment on the potential influence of surface roughness, and how it seems to vary for the different samples?

Response: We have not measured the surface roughness of the thin films. This is because; it cannot be precisely controlled and will vary film to film.

Increasing the surface roughness can increase TOF, because we can consider that film as isolated crystals. Please see the following references, which are also cited in the manuscript. We have compared our results with these two works.

<https://onlinelibrary.wiley.com/doi/full/10.1002/anie.202115100>

<https://pubs.acs.org/doi/10.1021/acscatal.0c00550>

6. On a similar note, there should be a flow rate during which film thickness should not influence the catalytic results, would it be possible to estimate this? This is perhaps similar to question number 9 below.

Response: If the reactants flow rate is fast enough that the reactants does not get enough time to diffuse into the pores of the thin film catalyst, in that case film thickness cannot influence the TOF. In this case size selectivity will be also negligible, because reactants cannot diffuse into the pores.

In the current setup of the cross flow, we could only reach up to 15 mL/min. At this point we do see thickness dependence.

It will be possible to measure the maximum flow, in future, by improving the design of the reaction setup. At this point our aim is to show the impact of diffusion control achieved by flow control. Hence, we did not pursue this at this point.

We would also like to note that the thickness independent flow rate is specific to the system, and cannot be generalized. Microfluidic volume, pore size and reactant size, all these factors will be important.

7. In Figure 1 and the related part of the introduction, are there any appropriate references that could be given for the fundamental relationship between diffusion length, TOF and selectivity? The caption of figure one also gives two proportionalities for how TOF varies with respect to diffusion length and flow rate. Are these established relationships?

Response: Thank you for this point. The Figure 1 is described based on fundamental laws of diffusion. We have cited the relevant papers 56-58.

We consider the turnover frequency (TOF) to be proportional to mass diffusion, assuming that the reaction occurs very rapidly. As a result, TOF depends on the diffusion length, following a quadratic relationship according to Fick's Law. For a spherical model, diffusivity is related to the radius by a power law with an exponent of 3.

Citations have been added in the appropriate sections.

8. It seems like an axis label is missing on Figure 2c, what is the x-axis?

Response: In Figure 2c x-axis defines two different catalysis set ups, one-pot and thin film. We have revised the figure accordingly.

9. What is the diffusion length under the studied conditions? Is there a point where the film thickness becomes irrelevant (i.e. diffusion into film does not occur past a certain point with a certain flow rate?)?

Response: This point is same as point 6. The response is as follows:

If the reactants flow rate is fast enough that the reactants does not get enough time to diffuse into the pores of the thin film catalyst, in that case film thickness cannot influence the TOF. In this case size selectivity will be also negligible, because reactants cannot diffuse into the pores.

In the current setup of the cross flow, we could only reach up to 15 mL/min. At this point we do see thickness dependence.

It will be possible to measure the maximum flow, in future, by improving the design of the reaction setup. At this point our aim is to show the impact of diffusion control achieved by flow control. Hence, we did not pursue this at this point.

We would also like to note that the thickness independent flow rate is specific to the system, and cannot be generalized. Microfluidic volume, pore size and reactant size, all these factors will be important.

10. What is the expected diffusion rate of the substrates inside the MOF? As I understand it, the argument is that a thicker film would give a longer diffusion length (L_d) when compared to a thinner film (assuming the same flow rate), why is that the case? This perhaps is closely related to question number 8 above.

Response: The diffusion rates of the substrates are predicted using MD simulation. This is discussed in the manuscript and highlighted in the revised version. The difference in the diffusivities is 10x between the large and small reactants.

We did not measure/calculate the absolute diffusivities of the reactants. This is because we realized that absolute values do not affect the observed results. Rather it is important to estimate the differences in diffusivity to correlate with the observed size selectivity.

Regarding the second part of the question; diffusion length is same as film thickness. This point is discussed in in point number 3.

11. Considering the pore aperture size of the framework (UiO-66-NH₂), would the authors expect the substrates to fit into the pores of the framework? How does the size

of the pore aperture compare to the bulkiest groups of the substrates that are used? Can the substrates be expected to reach the interior of the film?

Response: The pore and cavity sizes of the framework are 6 and 11 Å. The longest axis of the bulkiest substrate (tert-butyl cyanoacetate) is 10.3 Å. This size fitting discussion is included in the manuscript and highlighted in yellow.

We have performed the molecular dynamic simulation of the substrates included in the MOF pore. As can be seen from the Figure S8, the substrates can easily fit in the pores of UiO. We have also provided the structure file of the reactant included MOF structure in the revised supporting information.

12. A common reason for enhanced catalytic activity in UiO-66 is the presence of various defects. Could the authors comment on any potential presence of defects in their materials? Particularly how their synthesis route would compare to typical syntheses of more or less defective Ui-66-NH₂? As an example, a range of missing-linker defect concentrations are seen in materials prepared with varied linker-to-modulator ratios (10.1021/acs.chemmater.6b00602, 10.1016/j.cej.2023.143394). Could this be a potential confounder for the observed differences in catalytic performance of their materials or thin films?

Response: Thank you for this question.

The UiO-66-NH₂ thin films and polycrystalline powder catalysts used in this work is not defect free. The different sized crystals are made by varying modulator concentration and this will lead to varying amount of linkers. This is not avoidable. The synthesis methodology is adopted from the earlier investigations from our group and Hupp and coworkers.

It is very likely that defects do play a role in the determined TOF and selectivity in Figure 3a. However, Figure 3b is not influenced by defects.

In Figure 3b we conclude that TOF and selectivity is controlled by flow rate.

For Figure 3a, i.e. thin films of different thicknesses, we have now included comparative IR, XPS and XRD (see below). We have observed very similar defect densities. This is elaborated in the next point, as the question asked seems to be very similar.

These points are now discussed in the manuscript and highlighted in yellow.

This should at the very least be critically discussed considering that the differences in catalytic activity is the main finding of the paper. One approach for deducing this, at least for the powdered samples, would be thermogravimetric analysis, showing the

lower/higher mass loss as the linker is burned off in air (10.1021/acs.chemmater.6b00602). The current evidence presented by the authors, using XPS, is only sensitive of the nature of the exterior surface, and gives little information about the composition of the rest of the film. Even then, doing a quantitative analysis of XPS data requires the use of relevant standards as a comparison. The authors currently only discuss the composition of their materials as a comparison between particles vs. films, I believe this discussion of composition and defects needs to be extended to the films of various thicknesses (or particles of different sizes), as their defect concentrations may be radically different (and thereby giving rise to the differences in catalytic performance).

Response: Thank you for this comment. We understand that defects density (missing linker/node) can be a critical parameter for catalysis. It is very obvious that in the studied system, structural defects are present. However, we ascertain (based on experimental evidences) that defect densities do not control the final hypothesis of the paper, i.e. diffusion controlled selectivity. A smoking gun evidence is flow rate dependent TOF and selectivity (Figure 3b), for the same thin film.

As suggested by the referee, we have rechecked and compared the results to quantify defects in different samples. We have performed a comparison study using IR, XPS and XRD. All of these are included in the revised version as Figure S24, 25, 17, respectively.

IR shows nearly identical COO stretching frequency intensities, corresponding to noncoordinated and coordinated modes, for the different thin films and different powder MOF catalysts (Figure S24).

High resolution XPS of the thin films also indicate indifferent nature of the C, Zr, O and N (Figure S25). Although XPS is more sensitive to surface, very similar binding energies reiterate that the thin film defect densities are very similar. Different thickness (i.e. diffusion length) is the only reason for different TOF and selectivity (Figure 3a).

Figure S17 shows very similar crystallite domain sizes for the three different thin films.

All these evidences put together, we believe that the only defect density controlled TOF and selectivity can be ruled out.

See the figures appended below.

Figure S24: ATR and IRRA spectra of the submicron sized UiO-66-NH₂ crystallites and UiO_{TF1}-

Figure S25: XPS comparison of the UiO_{TF1-3} and 160 nm crystallites of UiO-66-NH₂. a) overview scan, b-e) high-resolution scan of C, Zr, O, and N respectively.

Figure S17: XRD patterns of the UiO_{TF1-3}, showing consistent full-width half maximum (FWHM) across all the three samples.

13. It would be nice to show that the thin films remain crystalline after the catalysis - they likely are, as Zr-MOFs are used for their high chemical robustness - but PXRD patterns taken also after catalysis should be provided.

Response: We have performed the XRD experiment after catalysis and included that in the revised supporting information (Figure S16).

Figure S16: XRD patterns of the different thicknesses of UiO-66-NH₂ thin films.

14. On another note regarding PXRD measurements, is the instrument listed correct? I can not seem to find records of Rigaku producing a PXRD instrument called "XDS 2000". Perhaps the manufacturer has been mixed up? Also, 200 mA seems to be a very high current for a conventional X-ray tube, but it may very well be correct, I would urge the authors to double-check.

Response: Thank you for this correction. Instrument name is Rigaku Smartlab. 200 mA current is correct. We have corrected this in the revised version.

Point-by-point response

Reviewer #1

The authors have done a very good job in addressing the numerous issues raised during review. The manuscript has significantly been strengthened by the new data/information added to the Supporting Information and by some specific sections/sentences to the main text / discussion, as explained in the response letter. The revisions are conclusive. I am happy to recommend acceptance of the manuscript as it is now.

Response: Thank you for recommending its publication.

Reviewer #2

The authors have certainly made a great effort to address comments from the reviewers. I am grateful for the thorough explanations provided by the authors, in particular regarding the mechanistic understanding of the selectivity and diffusion processes, as well as the description of the diffusion length. After reading the response letter and the revised manuscript, I have listed my comments and questions below, which I believe should be addressed prior to acceptance. Most pressing is perhaps the proof of the substrates fitting into the pore apertures of the porous catalyst. While the QCM results give new insights, I believe the discussion would benefit from the results being put into context (i.e. how the uptake relates to the amount of catalyst).

Response: Thank you for the suggestions. In the following we have addressed the concerns in point-by-point manner.

1. In their second response to reviewer one, the authors state that the defect density has been accounted for. This is mainly inferred through XPS and IR data, yet a more definitive and quantitative measure would be any deviation in the stoichiometry of the materials as determined by e.g. thermogravimetric analysis. While not possible to carry out on the thin films, could the authors perhaps comment on the relative ratios between

the different oxygen environments (Zr-O-Zr, Zr-O-C, COOH) for the different thin films as observed in the XPS results? Visually it appears to be similar between the thin films, as the authors highlight in the main text.

Response: Thank you for this suggestion. The relative ratios are now included in the revised supporting information (Figure S25f). The differences are very small, as mentioned previously.

2. Additionally, regarding the XPS results and the defect concentration, the C-C and C-N binding energies differ for the thin films, and a handful of peaks in the XPS data are shifted when comparing the thin films vs. the small particle sample. Could the authors comment on what the possible differences are between the thin films and the small particles? Are the magnitudes of these shifts worth considering?

Response: For the small particles and thin films, the differences in the specific binding energies are small, <0.1 eV. The differences can be only due to surface nature of the thin films and particles. Hence, we are unable to attribute these small changes to any specific differences in the defect densities.

3. Regarding the new QCM results, would it be possible to give the numbers in specific uptake (ng/g_MOF or similar)? Could the authors possibly discuss the magnitude of the uptake? Also, at what temperature was the measurement carried out? Would it be possible to measure a blank sample to make sure that the uptake is solely due to the porous material? Could the uptake be occurring only at the surface for the tert-butanol case?

Response: Thank you for this suggestions.

We have modified Figure S18 according to the suggestion. The measurement was carried out at 298 K.

The adsorption is due to the porous material. This is proved by the performing the suggested experiment. The blank QCM sensor showed negligible change in frequency,

compared to the MOF coated sensor. See the comparison profiles appended below (Figure R1).

Figure R1: Tertiary butanol vapor uptake profile using UiO-66-NH₂ MOF coated sensor and uncoated Au-sensor at 298 K.

4. The MD simulations show displacements that are quite small. If I interpret the y-axis values correctly, the movement is only within a single pore. If this is correct, I do not believe it shows the feasibility of the bulkier substrates to pass through the pore aperture. Would it be possible to carry out simulations at longer timescales to show diffusion between pores? On this note, I agree that the substrates fit inside the pores, but I am more hesitant if they can pass through the aperture, which, when taking into account the van der Waals radii of the carbon atoms, has a limiting size of about $\sim 4 \text{ \AA}$ depending on the orientation of the aromatic rings on the organic linker molecule, and further assuming that all amine groups are pointed away from the pore.

Response: We thank the reviewer for the insightful comment and agree with the observations regarding the movement of molecules within a single pore. Indeed, the slow diffusion of the molecules necessitates extremely long simulation times to observe

molecular transport between pores through unbiased molecular dynamics (MD) simulations. To address these computational limitations, we have employed biased simulation techniques, specifically umbrella sampling, to estimate the free energy barrier associated with molecular transport across the MOF pore window.

Figure R2: Figure shows the collective variable chosen for umbrella sampling simulation involving molecular movement from aperture of one pore to another for *Et-CA* (represented in cpk representation). The same is considered for the other analyte i.e. *t-But-CA* also.

Figure R3: Free energy profile corresponding to molecular transport through UiO-66-NH₂ pores compared for *Et-CA* and *t-but-CA*.

This approach not only mitigates the computational constraints of extended simulations but also allows for a quantitative comparison of the free energy barriers for the analytes *Et-CA* and *t-But-CA*. The umbrella sampling simulations were conducted to explore the free energy landscape associated with molecular movement from aperture of one pore to another for each analyte individually (Figure R2). The results, presented in Figure R3, indicate that the free energy barrier for *t-But-CA* is higher than that for *Et-CA*, likely due to its larger size. This higher barrier suggests that *t-But-CA* diffuses more slowly across the pore aperture compared to *Et-CA*.

We have included the results in the revised manuscript and discussed as following: “Furthermore we have employed biased simulation techniques, specifically umbrella sampling, to estimate the free energy barrier associated with molecular transport across the MOF pore window. Figure S, indicates that the free energy barrier for *t-But-CA* is

higher than that for Et-CA, likely due to its larger size, suggesting slower diffusion of *t*-But-CA across the pore aperture compared to Et-CA.” These changes are highlighted in yellow.

In addition to MD simulation, to understand how the bulky *t*-But-CA does diffuse from pore to pore, we have estimated the pore size distribution (Figure S1b) and % of linker defect in the UiO-66-NH₂ MOF (Figure S24b). We have performed these experiments for MOF particles (synthesis method is described). From the above mentioned experiments we concluded presence of >1 nm pore and ~33% missing linker defect. The large voids are created by missing linker defects, and this additionally facilitates diffusion. We have added this discussion in the revised manuscript and highlighted in yellow.

5. Additionally, how were the distances defined for the CH₂-to-linker/cluster interactions? I.e. what atom on the cluster or linker was used? For the amino group distance (Figure S9c), was the H₂C - - - NH₂ distance used?

Response: We thank the reviewer for the comment. We would like to clarify that the term metal-node represents the Zr metal and the linked oxygen atoms, μ_3 -O and μ_3 -OH. Whereas the term linker represents all the atoms of the organic linker i.e. bdc-NH₂. We computed the pair correlation function (using “gmx rdf”) between the CH₂ group of the analyte with respect to each atom of the metal-node (or organic linker) followed by normalization. This is included in the revised manuscript and highlighted in yellow.

6. Also pertaining to the simulations (Figure S9), is it diffusion constant or diffusivity (diffusion coefficient)? Depending on the answer, is the unit correct? Are the authors certain about the number of significant digits in their results?

Response: We thank the reviewer for bringing this to our attention. Regarding the simulations in Figure S9, the term should indeed refer to the diffusion constant. In the revised manuscript, we have corrected the unit from nm² to nm²/s to ensure accuracy and consistency.

7. Lastly regarding the MD results, would it be possible to include the structure file in an electronic format rather than as lines in the supporting information? I believe this would

greatly improve the ease of access to the data, and also greatly reduce the number of pages of the document.

Response: We thank the reviewer for the suggestion. As per the suggestion in the revised manuscript we have removed the snapshots from the supporting documents and provided the structure file for the ease of access of the data.

8. Regarding the SEM results, some of the images is acquired at 5 kV and 25 kV, not only at 30 kV as is listed in the Methods section.

Response: Thank you for this correction. We have corrected this and highlighted the modification in yellow.

9. At line 118 the authors state that “We anticipated the reaction rate to exceed the diffusion rate”. Does this mean, that under the studied conditions, the reaction rate is high enough so that the diffusion length does not become limiting?

Response: Sorry for not making this sentence clear.

We expected that the reaction rate for the condensation to be high, so that the slow diffusion of the substrates controls the TOF and size-selectivity.

10. This is merely this reviewer being picky, but perhaps “in heterogeneous catalysts” at line 45 should be changed to “in porous heterogeneous catalysts”.

Response: Thank you for this correction. We have corrected this and highlighted the modification in yellow.

11. The authors also talk interchangeably about “monolith”, “thin film”, and monolithic thin film. Perhaps consistently using “monolithic thin film” would be preferable? I leave this to the discretion of the authors, I just wanted to raise the question.

Response: Thank you for this correction. We have corrected this as “monolithic thin film”. All the changes are highlighted in yellow.

12. Is there a reason why the NMR results are truncated to show shifts only between 8 to 10 ppm? Could the authors provide the full spectra and interpretation? Typically, a range from -1 to 9 ppm is recommended.

Response: 8-10 ppm region exhibits characteristic H-peaks (marked as Ha and Hb), using which the product conversion % is calculated. The full scale NMR spectra of the products formed are shown in Figure S5 and 6. By showing specific region in the NMR, we wanted to show the product conversion % with more clarity.

For reference, each of the raw NMR data will be made available in the final version of the manuscript.